# Colorectal Cancer Prognosis Is Not Associated with *BRAF* and *KRAS* Mutations-A STROBE Compliant Study

**DOI:** 10.3390/jcm8010111

**Published:** 2019-01-17

**Authors:** Joon-Hyop Lee, Jiyoung Ahn, Won Seo Park, Eun Kyung Choe, Eunyoung Kim, Rumi Shin, Seung Chul Heo, Sohee Jung, Kwangsoo Kim, Young Jun Chai, Heejoon Chae

**Affiliations:** 1Department of Surgery, Gachon University College of Medicine, Gil Medical Center, Incheon 21565, Korea; deftnovice@gmail.com; 2Division of Computer Science, Sookmyung Women′s University, Seoul 04310, Korea; jiyoung464@naver.com; 3Department of Surgery, Kyung Hee University Hospital, Seoul 02447, Korea; pwsmd@hanmail.net; 4Department of Surgery, Seoul National University Hospital Healthcare System, Gangnam Center, 101 Daehak-ro, Seoul, Korea; choe523@gmail.com; 5Department of Surgery, National Medical Center, Seoul 04564, Korea; keyss30809@naver.com; 6Department of Surgery, Seoul Metropolitan Government-Seoul National University Boramae Medical Center, Seoul 07061, Korea; roomie79@gmail.com (R.S.); heosc3@brmh.org (S.C.H.); 7Division of Clinical Bioinformatics, Biomedical Research Institute, Seoul National University Hospital, Seoul 03080, Korea; sh0202j@gmail.com (S.J.); kksoo716@gmail.com (K.K.)

**Keywords:** colorectal cancer, *BRAF*, *KRAS*, overall survival, disease-free survival

## Abstract

Background: We investigated the associations between v-Raf murine sarcoma viral oncogene homolog B1 (*BRAF*^V600E^, henceforth *BRAF*) and v-Ki-ras2 Kirsten rat sarcoma viral oncogene homolog *(KRAS*) mutations and colorectal cancer (CRC) prognosis, using The Cancer Genome Atlas (TCGA) and the Gene Expression Omnibus (GSE39582) datasets. Materials and Methods: The effects of *BRAF* and *KRAS* mutations on overall survival (OS) and disease-free survival (DFS) of CRC were evaluated. Results: The mutational status of *BRAF* and *KRAS* genes was not associated with overall survival (OS) or DFS of the CRC patients drawn from the TCGA database. The 3-year OS and DFS rates of the *BRAF* mutation (+) vs. mutation (−) groups were 92.6% vs. 90.4% and 79.7% vs. 68.4%, respectively. The 3-year OS and DFS rates of the *KRAS* mutation (+) vs. mutation (−) groups were 90.4% vs. 90.5% and 65.3% vs. 73.5%, respectively. In stage II patients, however, the 3-year OS rate was lower in the *BRAF* mutation (+) group than in the mutation (−) group (85.5% vs. 97.7%, *p* < 0.001). The mutational status of *BRAF* genes of 497 CRC patients drawn from the GSE39582 database was not associated with OS or DFS. The 3-year OS and DFS rates of *BRAF* mutation (+) vs. mutation (−) groups were 75.7% vs. 78.9% and 73.6% vs. 71.1%, respectively. However, *KRAS* mutational status had an effect on 3-year OS rate (71.9% mutation (+) vs. 83% mutation (−), *p* = 0.05) and DFS rate (66.3% mutation (+) vs. 74.6% mutation (−), *p* = 0.013). Conclusions: We found no consistent association between the mutational status of *BRAF* nor *KRAS* and the OS and DFS of CRC patients from the TCGA and GSE39582 databases. Studies with longer-term records and larger patient numbers may be necessary to expound the influence of *BRAF* and *KRAS* mutations on the outcomes of CRC.

## 1. Introduction

Colorectal cancer (CRC) is one of the most common cancers worldwide, representing the second most common form of cancer diagnosed in females and the third most common form of cancer diagnosed in males [1]. Although the decreasing mortality rate is attributed to early detection and treatment advances, CRC currently ranks fourth for cancer-related mortality [2]. The TNM stage, a staging system defined by the American Joint Committee on Cancer (AJCC) based on pathologic and clinical factors, is the conventional parameter by which CRC prognosis and treatment is determined [3].

Research into the molecular and genetic mechanisms of CRC carcinogenesis and progression in recent decades prompted the search for genetic prognostic factors to complement the TNM staging system for all malignancies. Public availability of genomic databases such as The Cancer Genome Atlas (TCGA) and the Gene Expression Omnibus (GEO, GSE39582) dataset now allows clinicians and bioinformatics researchers to conduct genomic research on different cancers, including CRC, which has not generally been possible in the past [4,5,6,7]. There are several different molecular and genomic mechanisms that may contribute to CRC formation such as microsatellite instability phenotype, CpG island methylator phenotype [8], and chromosomal instability [9].

Research and systematic reviews have primarily focused on v-Raf murine sarcoma viral oncogene homolog B1 (*BRAF*) and v-Ki-ras2 Kirsten rat sarcoma viral oncogene homolog (*KRAS*) genes as potential prognostic biomarkers for CRC. Despite conflicting reported effects of *BRAF* and *KRAS* mutations [10,11,12,13] the presence of either are additional factors recommended in the 8th Edition of the AJCC guidelines for influencing clinical care [3,11,14]. In the present study we assessed the association between *BRAF*^V600E^ (*BRAF*) and *KRAS* mutations and the outcome of CRC patients by TNM stage using the publicly available TCGA and GSE39582 datasets.

## 2. Materials and Methods

### 2.1. Data Acquisition and Selection

The data of 536 patients with genomic variant colon adenocarcinoma (COAD) and rectal adenocarcinoma (READ) with corresponding clinical information were obtained from the TCGA GDC portal (https://portal.gdc.cancer.gov/). Their mRNA-Seq data were produced with the Illumina HiSeq 2000 (Illumina Inc, CA, USA) platform and processed by the RNAseqV2 pipeline, using MapSplice, v.1.15.2 (University of Kentucky, Lexington, KY, USA) and RSE, v. 1.3.0 for alignment and quantification. After excluding 89 patients with incomplete data (five with lack of clinical information, 69 missing survival data, and 15 lacking CRC stage information), the remaining 447 patients’ data were analyzed. In addition, the GSE39582 data of 585 patients were downloaded from the GEO repository (https://www.ncbi.nlm.nih.gov/geo/) [15]. Gene expression profiles were assessed with the Affymetrix U133 Plus 2.0 chip (Thermo Fischer Scientific, MA, USA). Eighty-eight patients were excluded because of incomplete data (40 missing *KRAS* mutational status, 35 missing *BRAF* mutational status, 4 missing tumor stage information, and 9 missing survival data). The remaining 497 patients’ data were analyzed in the same manner as the TCGA data. The median record length duration (henceforth follow-up period) was described in months with range information. CRC was staged according to the 8th American Joint Committee on Cancer’s TNM stages guidelines [3]. Sub-stages were omitted to simplify clinical data.

According to TCGA publication guidelines, there are no restrictions on the publication of these somatic mutation and mRNA sequencing data and no specific permission is required for investigators to publish papers containing or referring to these data (http://cancergenome.nih.gov/publications/publicationguidelines). Furthermore, because IRB approval is not acquired for use of public datasets, our study (which used the TCGA and GEO public databases) was not submitted for such review. The manuscript was written in accordance to the Strengthening the reporting of Observational Studies in Epidemiology protocol [16].

### 2.2. Definition of BRAF and KRAS Mutation Status

For the TCGA dataset, we categorized *BRAF* or *KRAS* mutations based on information provided by the Mutation Annotation Format files of the Mu-tect caller [17]. The mRNA-Seq data were generated using the Illumina HiSeq 2000 platform and processed by the RNAseqV2 pipeline. Patients were classified positive or negative by their mutation status. In the GSE39582 series, information on the gene expression profiles of the *BRAF* and *KRAS* genes was determined by the Affymetrix U133 Plus 2.0 chips. Those who were indicated as “M” were classified in the mutation (+) groups and those showing “WT” were categorized in the mutation (−) groups.

### 2.3. Kaplan–Meier Survival Analysis of BRAF and KRAS Mutations

The potential influence of *BRAF* and *KRAS* mutations on each TNM stage on 3-year patient overall survival (OS) and disease-free survival (DFS) was analyzed using the life table method. The TCGA and the GSE39582 dataset provided clinical information on OS and DFS. Kaplan-Meier estimates were calculated for all possible combination of *BRAF* and *KRAS* mutation over each TNM stage. The estimation was calculated by a survival package [18] and plotted by the ggplot2 package [19] in R (version 3.4.2) [20]. Null hypotheses of no difference were rejected if *p*-values were less than 0.05, or, equivalently, if the 95% confidence intervals of risk point estimates excluded 1.

## 3. Results

There were 627 CRC patients in the sample that we drew from the TCGA dataset, among which 180 were excluded from analysis due to missing values. From the remaining 447 patients, 83 (18.7%) were stage I, 172 (38.4%) were stage II, 134 (30.0%) were stage III, and 58 (12.8%) were stage IV. In the TCGA database, the median follow-up period of *BRAF* mutation (+) patients was 22.04 (0–133.55), for *BRAF* mutation (−) patients the median follow-up period was 23.98 (0–147.90), for *KRAS* mutation (+) patients it was 24.73 (0–140.28), and for *KRAS* mutation (−) patients it was 22.04 (0–147.90) months. There were 585 patient data in the GSE39582 dataset, among which 88 were excluded due to missing values. From the remaining 497 patients, 32 (6.4%) were stage I, 242 (48.7%) were stage II, 163 (32.8%), were stage III, and 60 (12.1%) were stage IV. The median follow-up period of the *BRAF* mutation (+) patients was 46 (1–172), for *BRAF* mutation (−) patients it was 52 (0–201), for *KRAS* mutation (+) patients it was 46 (0–201), and for *KRAS* mutation (−) patients it was 56 (0–192) months (Table 1).

### 3.1. Individual Effect of BRAF Mutation on Survival Outcome

In TCGA dataset, there was no significant effect of *BRAF* mutation on CRC OS and DFS. For the *BRAF* mutation (+) vs. mutation (−) groups, the 3-year overall survival (OS) rates were 92.6% vs. 90.4%, respectively (*p* = 0.4, number of patients: 13 vs. 109), and the 3-year disease-free survival (DFS) rates were 79.7% vs. 68.4% (*p* = 0.47, number of patients: 11 vs. 84), respectively (Figure 1). In stage II patients, the 3-year OS rate was lower in the *BRAF* mutation (+) group than in the mutation (−) group (85.5% vs. 97.7%, *p* < 0.001, number of patients: 7 vs. 53). There was no difference in the 3-year DFS rate of stage II based on the *BRAF* mutation status (73.5% vs. 79.6% for *BRAF* mutation (+) vs. mutation (−) groups, *p* = 0.071, number of patients: 6 vs. 45) (Figure 2). There were no differences in the 3-year OS or DFS of stage I, III, or IV patients.

In the GSE39582 dataset, *BRAF* mutation status was not associated with the 3-year OS rate (75.7% mutation (+) vs. 78.9%, mutation (−) *p* = 0.6, number of patients: 30 vs. 300) or DFS rate (73.6% mutation (−) vs. 71.1% mutation (−), *p* = 0.79, number of patients: 27 vs. 252) of all CRC patients (Figure 1). The association between *BRAF* mutation status and 3-year OS for each group from stage I to IV was not significant (data not shown). Regarding 3-year DFS, only the stage III *BRAF* mutation (+) group (84.4%) demonstrated a higher 3-year DFS, compared to the mutation (−) group (62.4%, *p* = 0.046, number of patients: 15 vs. 75) (Figure 3).

### 3.2. Individual Effect of KRAS Mutation on Survival Outcome

In the TCGA dataset, there was no significant association between *KRAS* mutation status and CRC OS or DFS. For *KRAS* mutation (+) vs. mutation (−) groups, the 3-year OS rates were 90.4% vs. 90.5% (*p* = 0.64, number of patients: 61 vs. 61) and the 3-year DFS rates were 65.3% vs. 73.5% (*p* = 0.21, number of patients: 48 vs. 47), respectively (Figure 4). Likewise, the *KRAS* mutation status was not associated with significant differences across all stages for 3-year OS rates as well as DFS rates (data not shown).

In the GSE39582 dataset, *KRAS* mutation positivity had a detrimental effect on 3-year OS rate (71.9% mutation (+) vs. 83% mutation (−), *p* = 0.05, number of patients: 116 vs. 214) and DFS rate (66.3% mutation (+) vs. 74.6% mutation (−), *p* = 0.013, number of patients: 98 vs. 181) (Figure 4). Among the CRC stages, stage III patients with *KRAS* mutation exhibited a lower 3-year OS (71.9% mutation (+) vs. 89.4% mutation (−), *p* = 0.02, number of patients: 41 vs. 71) and DFS (54.0% mutation (+) vs. 73.5% mutation (−), *p* = 0.022, number of patients: 31 vs. 59) rates (Figure 5).

### 3.3. Combined Effect of BRAF and KRAS on Survival Outcome

The 3-year OS rate of both *BRAF* and *KRAS* mutation (+) patients was 57.1%, of both *BRAF* and *KRAS* mutation (−) patients it was 88.5%, of *BRAF* mutation (−) and *KRAS* mutation (+) patients it was 92.1%, and for *BRAF* mutation (+) and *KRAS* mutation (−) patients it was 100% (*p* < 0.01, number of patients: 2, 59, 11, and 50, respectively). There were no statistically significant differences regarding the 3-year DFS rate for the same gene combination (*p* = 0.27, number of patients: 2, 46, 9, and 38, respectively).

Further subgrouping by stage revealed a combined effect of *BRAF* and *KRAS* mutation in stage II CRC patients only. Among the stage II CRC patients, the 3-year OS rate of both *BRAF* and *KRAS* mutation (+) patients was 0%, of *BRAF* mutation (−) and *KRAS* mutation (+) patients 96%, of *BRAF* mutation (+) and *KRAS* mutation (−) patients it was 100%, and of both *BRAF* mutation (−) and *KRAS* mutation (−) 100% (*p* < 0.01, number of patients: 0, 33, 7, and 11 respectively). Among the stage II CRC patients, the 3-year DFS rate for both *BRAF* and *KRAS* mutation (+) patients was 25%, for *BRAF* mutation (–) and *KRAS* mutation (+) patients it was 76.3%, for *BRAF* mutation (+) and *KRAS* mutation (−) patients it was 83.1%, and for both *BRAF* and *KRAS* mutation (−) patients it wasc84.1% (*p* < 0.01, number of patients: 1, 28, 6, and 17, respectively). There were no statistically significant differences caused by the combined effect of *BRAF* and *KRAS* on CRC stages I, III, or IV, regarding 3-year OS (*p* = 0.71, 0.9, and 0.91, respectively) or DFS rates (*p* = 0.85, 0.44, and 0.61, respectively).

There were no patients with both *BRAF* and *KRAS* mutations in the GSE39582 dataset.

## 4. Discussion

This study did not find a consistent association between the 3-year OS and DFS rates of CRC patients and *BRAF* or *KRAS* mutation. Likewise, our subgroup analysis did not demonstrate consistent effect of *BRAF* or *KRAS* mutations on OS and DFS according to each TNM stage. At present, the TNM staging system is the only classification routinely used in clinical practice to prognosticate CRC outcome [3]. However, the AJCC’s 8th Edition of this classification system recommends *BRAF* and *KRAS* for consideration in clinical care [3]. Accounting for *BRAF* and *KRAS* applied concurrently with the TNM staging system was shown to improve the prognostic ability from a 0.61–0.68 to 0.63–0.71 concordance index (a performance measurement in survival analysis) for stage II and III CRC patients [21]. On the other hand, our results did not demonstrate a significant relationship between CRC outcome and *BRAF* or *KRAS* mutations. Although a decreased 3-year OS and DFS for *KRAS*-positive patients was observed in the GSE39582 database, its significance was circumscribed by the fact that the results were not reproduced by the counterpart TCGA database. One reason for the inconsistency is the discrepancy in the distribution of *KRAS* mutation between the two databases (384 (85.9%) vs. 197 (39.6%) for TCGA and GSE39582, respectively).

*BRAF* is a serine/threonine kinase that contributes to cell proliferation, survival, and differentiation [22]. The glutamate for valine substitution at codon 600 in exon 15 (V600E) is the most common *BRAF* mutation, and is found in more than 90% of human malignancies such as papillary thyroid cancer, ovarian cancer, melanoma and CRC [22,23,24]. Around 4.7 to 20% of CRC patients have a *BRAF* mutation [22]. In a recent study of 1049 CRC patients with known *BRAF* status, *BRAF* mutation had a detrimental effect on DFS and OS for stage I, II, and III CRC patients [25]. There is an association between *BRAF* status and poorer clinical outcomes, especially in advanced stage CRC [26,27,28]. However, results in a pooled study of three trials (FOCUS, COIN, and PICCOLO), were inconsistent, showing worse OS but similar DFS for *BRAF* mutation (+) patients compared to the *BRAF* mutation (−) counterparts [29]. Therefore, further research is required to elucidate the role of *BRAF* in CRC.

The *KRAS*, a proto-oncogene, is a member of the Ras subfamily. It activates signaling cascades including the mitogen-activated protein kinase and PI3K pathways that regulate proliferation, motility, differentiation, and survival [30]. Mutations of *KRAS* are found in 30–50% of CRCs and they are usually point mutations, primarily in codons 12 and 13 [31]. In metastatic CRC, *KRAS* mutation is predictive for anti-EGFR monoclonal antibodies such as cetuximab [32]. Although controversial, several meta-analyses reveal little prognostic value of *KRAS* mutations in CRC, in accordance with our results [14,33] (Table 2).

Co-existing *BRAF* and *KRAS* mutation are extremely rare because they are mutually exclusive [34]. In the unusual cases in which they do occur simultaneously, they seem to act synergistically to further shorten survival. In a study with 1000 CRC patients with known *BRAF* and *KRAS* status, only 3 (0.3%) harbored both mutations simultaneously and those patients fared significantly worse in terms of OS and DFS compared to their wild type counterpart [25]. In the GSE39582 dataset used in the present study, no patients had both mutations. Of the TCGA dataset, nine patients (2%) out of 447 simultaneously had both *BRAF* and *KRAS* mutations. Their OS was shorter than those with other mutational combinations. We found no significant difference in DFS among these nine patients, but this may be attributed to the small sample size and lack of longitudinal data. Although the biologic mechanism through which the two mutually exclusive mutations co-exist is not clear, tumor heterogeneity is one theory. A sole malignancy consists of many sub-clone tumor cells, which may explain how two mutually exclusive *BRAF* and *KRAS* gene mutations can co-exist in a single tumor. The clinical significance of such a rare occurrence should be noted regarding the prognosis of the CRC and investigated further.

A limitation of our study was the brevity of patient survival information; in the TCGA and GSE39582 dataset it was insufficient in certain subgroups to comprehensively assess survival. Hence, we acquired data on the 3-year, instead of 5-year, survival rate. According to the surveillance, epidemiology, and end results data of 28,491 cases of CRC, an approximate 10-point reduction in OS rate was observed between the third and fifth post-operative year [35]. Three years was long enough to stratify the survival rates by the TNM stage. Therefore, the 3-year rate was deemed a sufficient surrogate indicator for our study. Furthermore, the quality of additional information on post-operative chemotherapy was poor. Although whether the patients received palliative or adjuvant therapy was mentioned, the specific regimen, total dosage, and the number of sessions administered was not mentioned, therefore rendering further qualitative analysis impossible.

In conclusion, we were unable to exclude the role of chance in the association between the mutational status of either *BRAF* or *KRAS* on the survival outcomes of CRC patients. The definitive role of *BRAF* and *KRAS* mutations on CRC survival with longer-term observation remains to be fully described in studies.

## Figures and Tables

**Figure 1 jcm-08-00111-f001:**
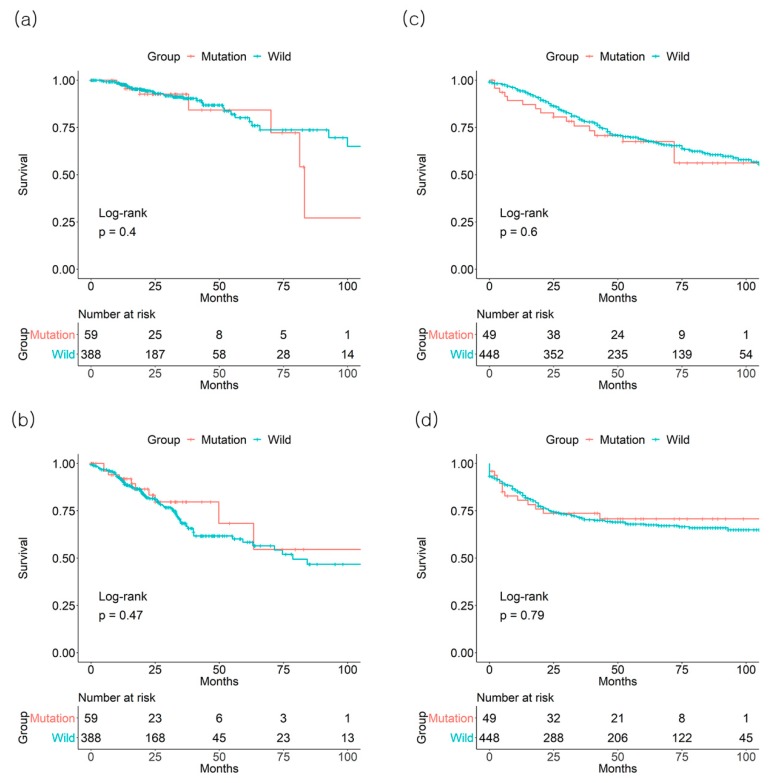
Effect of *BRAF* mutation on (**a**) 3-year overall survival and (**b**) 3-year disease-free survival from the The Cancer Genome Atlas (TCGA) database and on (**c**) 3-year overall survival and (**d**) 3-year disease-free survival from the Gene Expression Omnibus (GSE39582) database of colorectal cancer patients of all stages.

**Figure 2 jcm-08-00111-f002:**
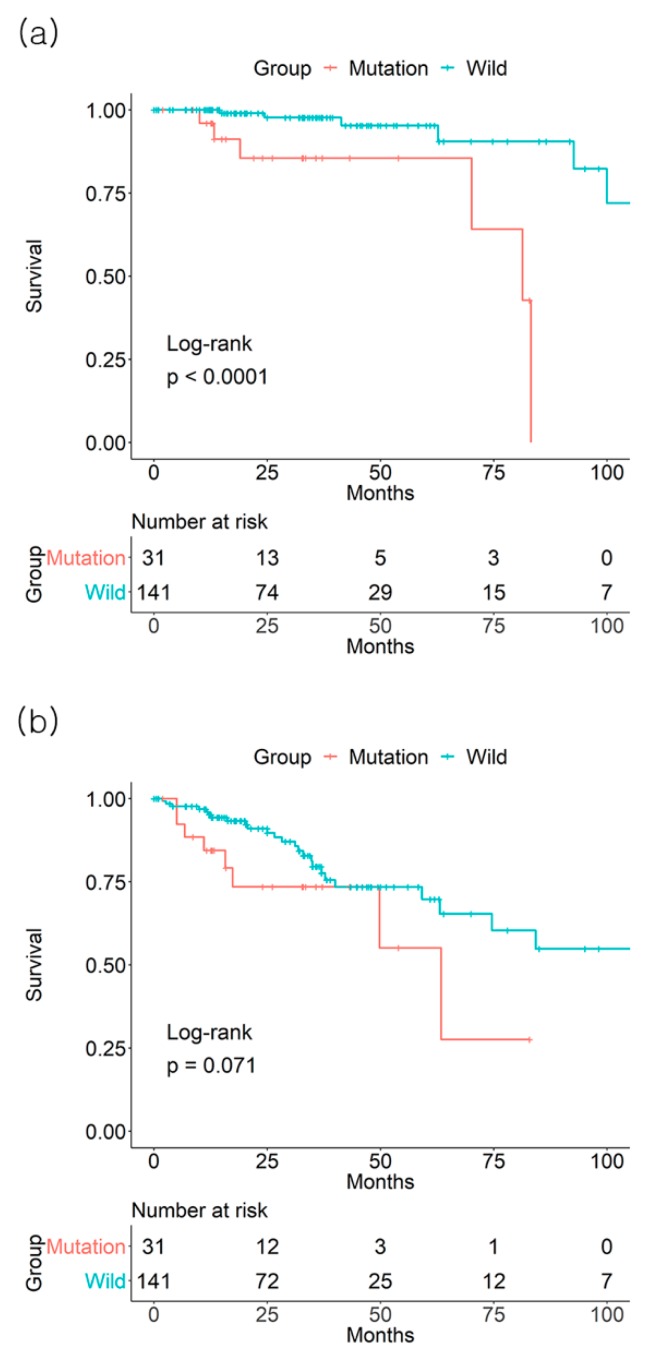
Effect of *BRAF* mutation on (**a**) 3-year overall survival and (**b**) 3-year disease-free survival from the The Cancer Genome Atlas (TCGA) database of colorectal cancer stage II patients.

**Figure 3 jcm-08-00111-f003:**
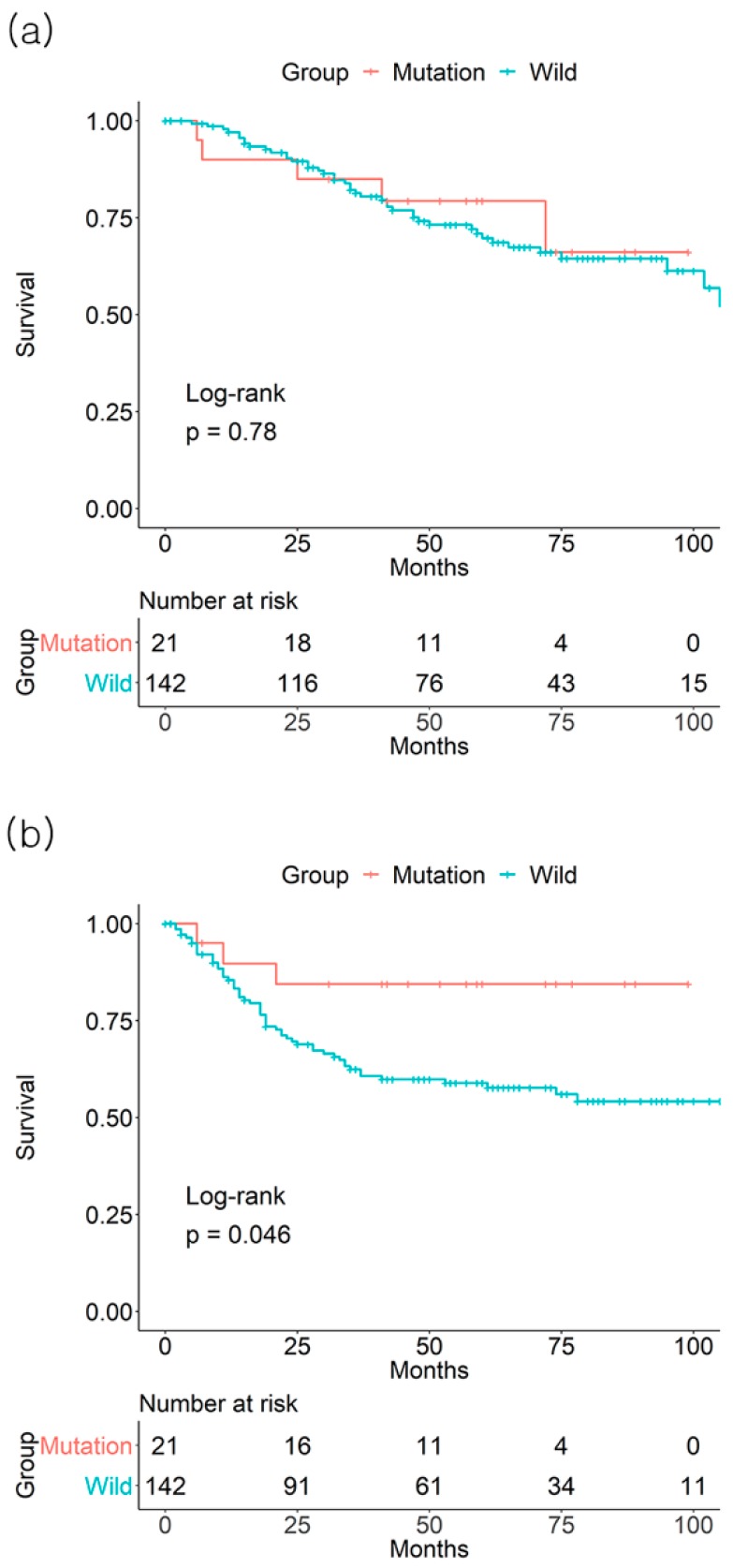
Effect of *BRAF* mutation on (**a**) 3-year overall survival and (**b**) 3-year disease-free survival from the GSE39582 database of colorectal cancer stage III patients.

**Figure 4 jcm-08-00111-f004:**
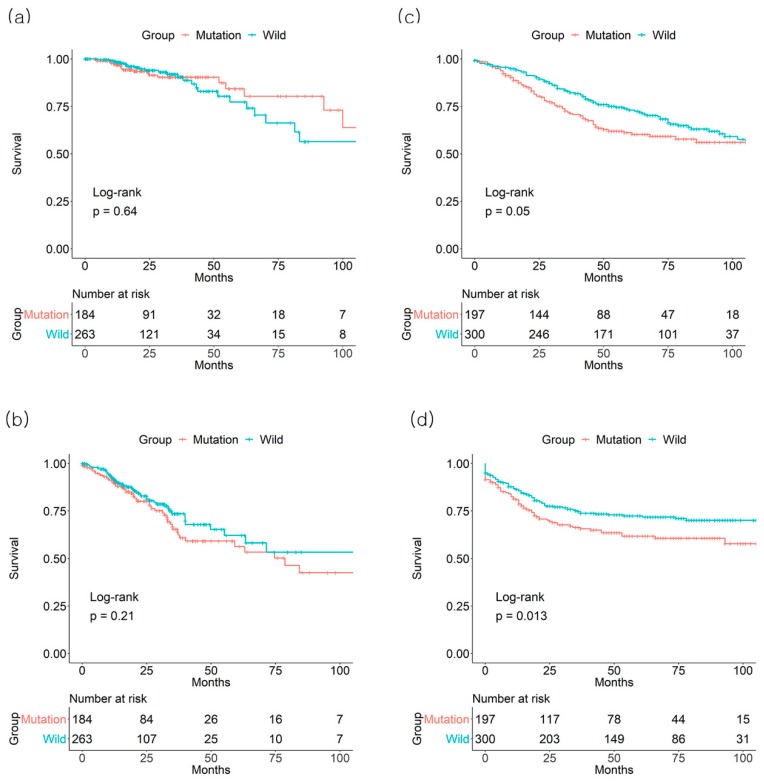
Effect of *KRAS* mutation on (**a**) 3-year overall survival and (**b**) 3-year disease-free survival from the The Cancer Genome Atlas (TCGA) database and on (**c**) 3-year overall survival and (**d**) 3-year disease-free survival from the GSE39582 database of colorectal cancer patients of all stages.

**Figure 5 jcm-08-00111-f005:**
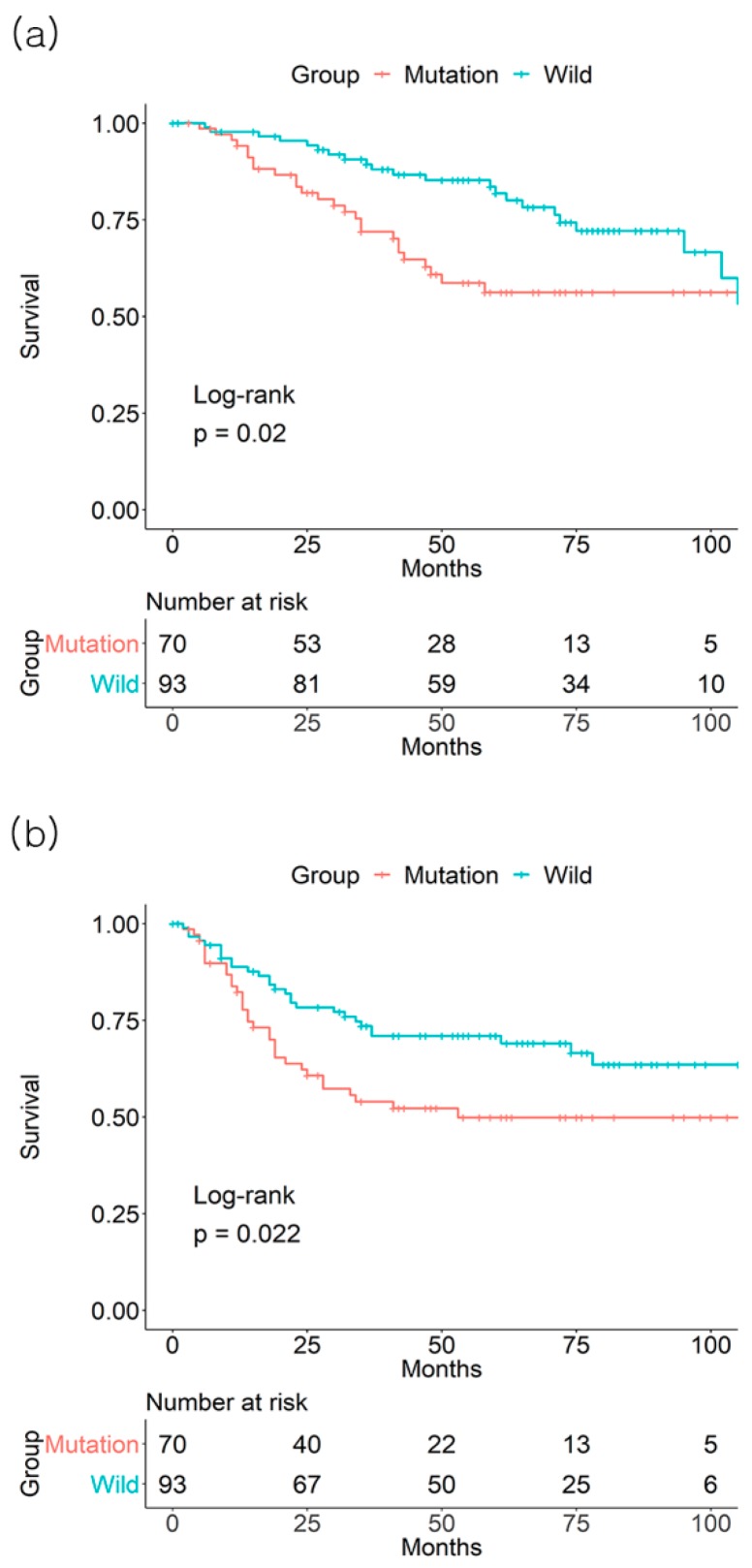
Effect of *KRAS* mutation on (**a**) 3-year overall survival and (**b**) 3-year disease-free survival from the GSE39582 database of colorectal cancer stage III patients.

**Table 1 jcm-08-00111-t001:** Baseline characteristics of datasets.

		TCGA	GSE39582
Total number of patients studied		447	497
Age at diagnosis (mean ± SD), years old		65.0 ± 12.9	66.8 ± 13.1
Gender			
	Male	213 (47.7%)	271 (54.5%)
	Female	234 (52.3%)	226 (45.5%)
Number of *BRAF* mutations		59 (13.2%)	49 (9.9%)
Number of *KRAS* mutations		384 (85.9%)	197 (39.6%)
AJCC TNM stage (number of *BRAF*/*KRAS* mutations)			
	I	83 (10/36)	32 (1/15)
	II	172 (31/77)	242 (21/83)
	III	134 (14/50)	163 (21/70)
	IV	58 (4/21)	60 (6/29)
Survival event			
	Dead	46 (10.3%)	40 (8.0%)
	Alive	404 (90.4%)	457 (92.0%)
Median follow-up time, months (range)		23.9 (0–147.9)	52 (0–201)
Median time to survival event, months (range)		24.4 (4.0–99.9)	31 (0–183)

TCGA: The Cancer Genome Atlas; AJCC: American Joint Committee on Cancer; SD: standard deviation; *BRAF*: v-Raf murine sarcoma viral oncogene homolog B1; *KRAS*: v-Ki-ras2 Kirsten rat sarcoma viral oncogene homolog.

**Table 2 jcm-08-00111-t002:** *BRAF*/*KRAS* status and prognosis in previous studies.

Author, Year	Country	Time Frame	Patients	Stage	Outcomes	Poor Prognosis
*BRAF*						
Won et al. 2017	South Korea	2010–2013	1049	I, II, III	OS *, DFS **	Mutant type
Seligmann et al., 2017	United Kingdom	2000–2003	231	IV	PFS ***	Mutant type
Roth et al. 2010	Switzerland	(−)	1404	II, III	OS *, DFS **	Mutant type
*KRAS*						
De Roock et al. 2010	Belgium	2001–2008	229	IV	PFS ***	No difference
Bokemeyer et al. 2011	Germany	2006–2007	315	IV	PFS ***	Mutant type
Schwartzberg et al. 2014	Spain	2009–2011	278	IV	PFS ***, OS *	Mutant type

* OS: overall survival. ** DFS: disease-free survival. *** PFS: progression-free survival.

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
