# Peer review of "Colorectal Cancer Prognosis Is Not Associated with *BRAF* and *KRAS* Mutations-A STROBE Compliant Study"

_jcm, 2019, doi:10.3390/jcm8010111_

Round 1

Reviewer 1 Report

In this manuscript the authors reported the results of the significance of BRAF and KRAS mutations in colorectal cancer survival. The study was conducted with methodological rigor. The results obtained can be considered further topics on the problem. The mutational status of KRAS drawn from the GSE39582 database were associated with significant decrease in OS or DFS; this result deserves more consideration in the discussion. 

Author Response

In this manuscript the authors reported the results of the significance of BRAF and KRAS mutations in colorectal cancer survival. The study was conducted with methodological rigor. The results obtained can be considered further topics on the problem. The mutational status of KRAS drawn from the GSE39582 database were associated with significant decrease in OS or DFS; this result deserves more consideration in the discussion.  

Thank-you for your kind comment: we have added a portion in the first paragraph of the discussion elaborating about the different results retrieved from the two databases

Although a decreased 3-year OS and DFS for KRAS positive patients was observed in the GSE39582 database, its significance was circumscribed by the fact that the results were not reproduced by the counterpart TCGA database. One reason for the inconsistency is the discrepancy in the distribution of KRAS mutation between the two databases (384 (85.9%) vs. 197 (39.6%) for TCGA and GSE39582, respectively).

Reviewer 2 Report

This study investigated the associations between BRAF and KRAS mutations and colorectal cancer (CRC) prognosis using the Cancer Genome Atlas (TCGA) and the Gene Expression Omnibus (GSE39582) dataset. There have been many studies on this topic using different patient groups. The idea of this study is not new and the conclusion is also not impressive.

1.  The median follow-up period in TCGA and GSE39582 is 23.9 and 52 months, respectively. Figure 1 showed survival curve up to 150 and 200 months, which is not necessary and meaningless because the number at risk at BRAF mutation group is only 1 after 100 months of survival. The authors compared the difference of 3-year OS and DFS (36 months) in this study. I think the time period of all figures should be modified.

2.  Please add the patient number of KRAS and BRAF mutation in table 1, and also mention the patient number of KRAS and BRAF mutation in each stage. Because throughout the whole result, no actually patient number in each group was mentioned, only percentage is shown. I wonder the meaning of any statistical difference if the actual patient number in each group is very small.

3.  About the combination effect of KRAS and BRAF mutation, the actual patient number is not mentioned at all. Please descript the patient number in each group in the result section 3.3. Because the author made a stage II subgroup analysis, the remaining patient number in this analysis must be very low. How could the author get a meaningful difference when the case number is extremely low?

Author Response

This study investigated the associations between BRAF and KRAS mutations and colorectal cancer (CRC) prognosis using the Cancer Genome Atlas (TCGA) and the Gene Expression Omnibus (GSE39582) dataset. There have been many studies on this topic using different patient groups. The idea of this study is not new and the conclusion is also not impressive.

1.  The median follow-up period in TCGA and GSE39582 is 23.9 and 52 months, respectively. Figure 1 showed survival curve up to 150 and 200 months, which is not necessary and meaningless because the number at risk at BRAF mutation group is only 1 after 100 months of survival. The authors compared the difference of 3-year OS and DFS (36 months) in this study. I think the time period of all figures should be modified.

We understand your point and have adjusted all the graphs according to your recommendation to show the survival up to 100 months only.

2. Please add the patient number of KRAS and BRAF mutation in table 1, and also mention the patient number of KRAS and BRAF mutation in each stage. Because throughout the whole result, no actually patient number in each group was mentioned, only percentage is shown. I wonder the meaning of any statistical difference if the actual patient number in each group is very small.

Thank-you very much for mentioning a very important point. We have modified Table 1. as below. Also the number of patients demonstrating overall survival and disease free survival for each mutation, stage and dataset was added throughout the whole results section

TCGA

GSE39582

Total number   of patients studied

447

497

Age at   diagnosis (mean ± SD), years old

65.0 ± 12.9

66.8 ± 13.1

Gender

Male

213 (47.7%)

271 (54.5%)

Female

234 (52.3%)

226 (45.5%)

Number of   BRAF mutations

59 (13.2%)

49 (9.9%)

Number of   KRAS mutations

384 (85.9%)

197 (39.6%)

AJCC TNM   stage (number of BRAF/KRAS mutations)

I

83 (10/36)

32(1/15)

II

172(31/77)

242(21/83)

III

134(14/50)

163(21/70)

IV

58(4/21)

60(6/29)

Survival   event

Dead

46 (10.3%)

40 (8.0%)

Alive

404 (90.4%)

457 (92.0%)

Median   follow-up time, months (range)

23.9   (0-147.9)

52 (0-201)

Median time   to survival event, months (range)

24.4 (4.0 ~   99.9)

31 (0-183)

3.  About the combination effect of KRAS and BRAF mutation, the actual patient number is not mentioned at all. Please descript the patient number in each group in the result section 3.3. Because the author made a stage II subgroup analysis, the remaining patient number in this analysis must be very low. How could the author get a meaningful difference when the case number is extremely low?

As mentioned in the previous response, the number of surviving patients after 3 years was added throughout the results section. The number of patients was very low for concomitant BRAF and KRAS mutation (+) patients, which was expected due to their mutually exclusive nature. Furthermore, we acknowledge that a definitive conclusion cannot be derived from a small patient group. We tried, however, to point to a trend that warrants further attention. This is mentioned in the discussion section.

Although the biologic mechanism through which the two mutually exclusive mutations co-exist is not clear, tumor heterogeneity is one theory. A sole malignancy consists of many sub-clone tumor cells, which may explain how two mutually exclusive BRAF and KRAS gene mutations can co-exist in a single tumor. The clinical significance of such rare occurrence should be noted regarding the prognosis of the CRC and investigated further.

Reviewer 3 Report

It is a well-prepared manuscript with an interesting finding and elegant demonstration. The author provides sufficient background details and conclusions are supported and are well justified. I suggest accept this version.

Author Response

Thank-you very much for your encouraging comment.

We are glad that you found our results interesting.

Reviewer 4 Report

The study by Lee et al. is interesting, however, some concerns need to be addressed.

The title need to be clarified. "Colorectal cancer survival" should be changed. In addition, the verb has been missed.

Gene names should be consequently written in italics.

Citations should be corrected eg., line 53.

Table 1, datasets with missing values are not necessary to be shown in this table as they do not give any information. It is enough to explain it within the text of the manuscript. It would be more important to state in Table 1 the actual number of samples studied.

What I have found missed in the manuscript is the information on the number of samples with specific mutations in BRAF and/or KRAS.

Are there any data on potential chemotherapy of patients from whom sequencing data were obtained? It is quite important as mutations in these driver oncogenes can be acquired in response to tumor cell adaptation to therapy.

Author Response

The study by Lee et al. is interesting, however, some concerns need to be addressed.

The title need to be clarified. "Colorectal cancer survival" should be changed. In addition, the verb has been missed.

Thank-you for your comment. We changed our title as below:

Colorectal cancer prognosis is not associated with BRAF and KRAS mutations – STROBE compliant study

Gene names should be consequently written in italics.

We have noticed the mistake and corrections have been made throughout the text

Citations should be corrected eg., line 53.

There must have been an error with the endnote program. Corrections have been made.

Table 1, datasets with missing values are not necessary to be shown in this table as they do not give any information. It is enough to explain it within the text of the manuscript. It would be more important to state in Table 1 the actual number of samples studied.

What I have found missed in the manuscript is the information on the number of samples with specific mutations in BRAF and/or KRAS.

Thank-you very much for mentioning the important points. We have modified Table 1. as below. Also the number of patients demonstrating overall survival and disease free survival for each mutation, stage and dataset was added throughout the whole results section

TCGA

GSE39582

Total number of patients studied

447

497

Age at diagnosis (mean ± SD), years   old

65.0 ± 12.9

66.8 ± 13.1

Gender

Male

213 (47.7%)

271 (54.5%)

Female

234 (52.3%)

226 (45.5%)

Number of BRAF mutations

59 (13.2%)

49 (9.9%)

Number of KRAS mutations

384 (85.9%)

197 (39.6%)

AJCC TNM stage (number of BRAF/KRAS mutations)

I

83 (10/36)

32(1/15)

II

172(31/77)

242(21/83)

III

134(14/50)

163(21/70)

IV

58(4/21)

60(6/29)

Survival event

Dead

46 (10.3%)

40 (8.0%)

Alive

404 (90.4%)

457 (92.0%)

Median follow-up time, months (range)

23.9 (0-147.9)

52 (0-201)

Median time to survival event, months   (range)

24.4 (4.0 ~ 99.9)

31 (0-183)

Are there any data on potential chemotherapy of patients from whom sequencing data were obtained? It is quite important as mutations in these driver oncogenes can be acquired in response to tumor cell adaptation to therapy.

We understand the importance of information on chemotherapy, especially with regards to KRAS mutation. However, the datasets only contained information on the palliative, adjuvant nature of the treatment without mentioning the specific regimen, dosage, and frequency. Therefore a qualitative analysis was not possible to carry out.

Round 2

Reviewer 4 Report

All concerns have been addressed.